# Effects of Hydrocodone Overdose and Ceftriaxone on Astrocytic Glutamate Transporters and Glutamate Receptors, and Associated Signaling in Nucleus Accumbens as well as Locomotor Activity in C57/BL Mice

**DOI:** 10.3390/brainsci14040361

**Published:** 2024-04-05

**Authors:** Woonyen Wong, Youssef Sari

**Affiliations:** Department of Pharmacology and Experimental Therapeutics, College of Pharmacy and Pharmaceutical Sciences, The University of Toledo, Toledo, OH 43614, USA; woonyen.wong@rockets.utoledo.edu

**Keywords:** ceftriaxone, GLT-1, opioids, glutamate, xCT

## Abstract

Chronic opioid treatments dysregulate the glutamatergic system, inducing a hyperglutamatergic state in mesocorticolimbic brain regions. This study investigated the effects of exposure to hydrocodone overdose on locomotor activity, expression of target proteins related to the glutamatergic system, signaling kinases, and neuroinflammatory factors in the nucleus accumbens. The locomotor activity of mice was measured using the Comprehensive Laboratory Animal Monitoring System (CLAMS). CLAMS data showed that exposure to hydrocodone overdose increased locomotion activity in mice. This study tested ceftriaxone, known to upregulate major glutamate transporter 1 (GLT-1), in mice exposed to an overdose of hydrocodone. Thus, ceftriaxone normalized hydrocodone-induced hyperlocomotion activity in mice. Furthermore, exposure to hydrocodone overdose downregulated GLT-1, cystine/glutamate antiporter (xCT), and extracellular signal-regulated kinase activity (p-ERK/ERK) expression in the nucleus accumbens. However, exposure to an overdose of hydrocodone increased metabotropic glutamate receptor 5 (mGluR5), neuronal nitric oxide synthase activity (p-nNOS/nNOS), and receptor for advanced glycation end products (RAGE) expression in the nucleus accumbens. Importantly, ceftriaxone treatment attenuated hydrocodone-induced upregulation of mGluR5, p-nNOS/nNOS, and RAGE, as well as hydrocodone-induced downregulation of GLT-1, xCT, and p-ERK/ERK expression. These data demonstrated that exposure to hydrocodone overdose can cause dysregulation of the glutamatergic system, neuroinflammation, hyperlocomotion activity, and the potential therapeutic role of ceftriaxone in attenuating these effects.

## 1. Introduction

Chronic treatment with opioids alters several neurotransmitters, particularly glutamate, in the central and peripheral systems. Many opioids are used for the management of pain. Among these opioids, hydrocodone is a semisynthetic and potent opioid agonist with a high abuse potential similar to fentanyl and morphine. Along with other opioids, hydrocodone is among the most prescribed opioids in the United States to treat moderate to severe pain [1]. Our current study focused on investigating the effects of exposure to hydrocodone overdose on locomotor activity and the expression of target glutamate transporters and signaling in one of the brain regions of the reward circuit, such as the nucleus accumbens (NAc).

Repeated exposure to drugs of abuse is known to increase motor activity, leading to behavioral hypersensitivity. For example, cocaine, amphetamines, morphine, ethanol, and nicotine have been reported to cause behavioral hypersensitivity [2,3,4,5]. Behavioral hypersensitivity induced by drugs of abuse relies on neural sensitivity and neuroplasticity within the reward circuit of the brain [6]. Other studies have shown that exposure to opioids, including hydrocodone, induces locomotor sensitization [7,8]. Although a previous study from our laboratory showed that ceftriaxone, a beta-lactam antibiotic known to upregulate the major glutamate transporter GLT-1, attenuated reinstatement to hydrocodone using conditioned place preference [9], the effect of ceftriaxone in attenuating hydrocodone-induced locomotor sensitization is less studied. Thus, in this study, we used the Comprehensive Laboratory Animal Monitoring System (CLAMS) to determine locomotion activity in mice exposed to repeated doses of hydrocodone with a challenge at a higher dose, and ceftriaxone treatment by breaking the infrared beam using infrared photocell technology.

We further focused on investigating the glutamatergic system. Glutamate function is regulated by several transporters and receptors, including the metabotropic glutamate receptor 5 (mGluR5), glutamate transporter 1 (GLT-1), known to regulate the majority of extracellular glutamate, and cystine/glutamate antiporter (xCT), which regulates glutamate output from astrocytes [10,11]. mGluR5 is highly expressed in the NAc [12], and it is an important mediator of synaptic plasticity and excitatory neurotransmission [13,14]. Exposure to substances of abuse downregulates GLT-1 and xCT expression in several brain regions, and beta-lactams (e.g., ceftriaxone) attenuate these effects [15]. Indeed, a study from our laboratory indicated that ceftriaxone treatment attenuates hydrocodone-induced downregulation of GLT-1 and xCT expression in mesocorticolimbic brain regions, as well as hydrocodone-induced upregulation of mGluR5 expression in chronic hydrocodone-exposed mice [16]. Increased GLT-1 expression by beta-lactams can lead to a sustained reduction in extracellular glutamate concentrations in the NAc [17] and consequently reduce the activation of mGluR5 downstream pathways, thereby reducing neuroexcitotoxicity. Therefore, in this study, we tested whether the beta-lactam ceftriaxone attenuates the effects of exposure to hydrocodone overdose on the expression of GLT-1, xCT, and mGluR5 in the NAc. We further tested the effects of exposure to hydrocodone overdose in the signaling pathways involved in neuronal nitric oxide synthase (nNOS) activity, the extracellular signal-regulated kinases (ERK) signaling pathway, and the receptor for advanced glycation end products (RAGE) as a signaling inflammatory marker, and whether ceftriaxone attenuates these effects. Additionally, this study explored the modulatory effects of ceftriaxone against hydrocodone-induced hyperlocomotion activity using CLAMS.

## 2. Materials and Methods

### 2.1. Animal Use Approval

All experimental procedures were approved by the Institutional Animal Care and Use Committee (IACUC) at The University of Toledo, protocol number 400155 (approved 2 August 2022). This protocol follows the guidelines for the use of animals in research, as described in the National Institutes of Health’s Guide for the Care and Use of Laboratory Animals. Mice were intraperitoneal (i.p.) injected and handled with care to prevent any distress and minimize any pain. The mice were monitored every day throughout the study, particularly when hydrocodone treatment was performed, and this was to determine any potential health issues. The mice were euthanized using CO_2_ inhalation and decapitated at the end of the experiment.

### 2.2. Animal and Study Design

C57BL/6 mice were used in this study. These mice were reported to show robust differences in drug dependence-relevant behaviors, including locomotor sensitization to substance abuse [18,19]. Eight-week-old male C57BL/6 mice (Jackson Laboratory, Bar Harbor, ME, USA, 25–30 g) were grouped into three groups: (1) control group (*n* = 7–8); (2) hydrocodone group (*n* = 7–8); and (3) hydrocodone–ceftriaxone group (*n* = 7–8). The animals were housed in a room maintained at 21 °C with a 12/12 h light/dark cycle. Hydrocodone (Sigma-Aldrich, St. Louis, MO, USA) was dissolved in saline at 20 mg/kg and 40 mg/kg, and ceftriaxone (Pfizer, New York, NY, USA) was dissolved in saline at 200 mg/kg. For acclimatation purposes, mice were handled for three days prior to the start of the experiment. The control mice received an i.p. injection of saline (vehicle) every other day from Day 1 to Day 9. Groups 2 and 3 received hydrocodone (20 mg/kg, i.p.) on Days 1, 3, 5, and 7. On Day 9, mice received an overdose of hydrocodone (i.p.) at 40 mg/kg. In addition, group 3 received ceftriaxone (200 mg/kg, i.p.) on Days 5–9, and groups 1 and 2 received equivalent volume of vehicle saline (i.p.) on Days 5–9. On Day 7 of the drug treatment, the mice were placed in Minispec NMR, which is used to measure the lean and fat mass of mice (Figure 1). It is important to note that 40 mg/kg (i.p.) of hydrocodone was considered a sublethal dose since we found that 45 mg/kg (i.p.) of hydrocodone was lethal in the mice (*n* = 3). Thus, we have chosen to test 40 mg/kg (i.p.) of hydrocodone, which was considered as a higher and sublethal dose.

### 2.3. Comprehensive Laboratory Animal Monitoring System (CLAMS)

Mice were placed individually in a Comprehensive Laboratory Animal Monitoring system (CLAMS; Columbus Instruments, Columbus, OH, USA) and had access to free food and water from day 7 to day 11 (Figure 1). The mice were placed at room temperature under alternating 12 h light and 12 h dark cycles. After adaptation for one day, individual locomotor activity was detected using IR photocell technology.

### 2.4. Brain Tissue Extraction

The mice were sacrificed using CO_2_ inhalation euthanasia procedure on Day 12 (Figure 1). The brains were dissected out and frozen on dry ice and further stored at −80 °C. NAc (core and shell) was extracted using a cryostat machine (Leica CM1950, Leica, Deer Park, IL, USA). The NAc was selected using the Brain Mouse Atlas [20]. NAc samples were stored at −80 °C for determination of target proteins using a Western blot assay.

### 2.5. Western Blot Analyses

Protein expression of phospho-nNOS, nNOS, RAGE, phospho-ERK, ERK, xCT, GLT-1, mGluR5, and β-tubulin was determined in the NAc (core and shell) using a Western blot assay. NAc tissues from all groups were lysed using lysis buffer (50 mM Tris-HCl, 150 mM NaCl, 1 mM EDTA, 0.5% NP-40, 1% Triton, 0.1% SDS) with phosphatase and protease inhibitors. Quantification of the amount of protein was performed using a detergent-compatible protein assay (Bio-Rad, Hercules, CA, USA). Protein from each sample with equal amounts was mixed with laemmili dye and further loaded onto 10% Tris-glycerine gel for separation of loaded proteins using an electrophoresis apparatus. Separated proteins were transferred from the gels into a polyvinylidene difluoride (PVDF) membrane. The PVDF membranes were incubated in 5% fat-free milk in Tris-buffered saline with Tween 20 (TBST) for 30 min at room temperature. The membranes were further incubated overnight at 4 °C with primary antibodies: rabbit anti-phospho-ERK (1:1000, Abcam, Waltham, MA, USA, ab201015), rabbit anti-ERK (1:1000, Abcam, Waltham, MA, USA, ab17942), rabbit anti-Phospho-nNOS (1:1000, Abcam, Waltham, MA, USA, ab16650), rabbit anti-nNOS (1:1000, Abcam, Waltham, MA, USA, ab76067), rabbit anti-RAGE (1:1000, Abcam, Waltham, MA, USA, ab37647), rabbit anti-GLT-1 (1:5000, Abcam, Waltham, MA, USA, ab205248), rabbit anti-xCT (1:1000, Abcam, Waltham, MA, USA, ab125186), and rabbit anti-mGluR5 (1:1000, Abcam, Waltham, MA, USA, ab76316). We used mouse anti-β-tubulin (1:1000, BioLeagend, San Diego, CA, USA) as a control loading protein. The next day, the membranes were washed with TBST five times and incubated with the corresponding secondary antibody (1:4000) for 60 min. The membranes were washed with TBST and dried for further analysis. The membranes were then incubated in chemiluminescent reagents (Super Signal West Pico, Perce Inc., Appleton, WI, USA) for 1–2 min. The GeneSys imaging system (Syngene, Frederick, MD, USA) was used for blot development and digitization. The expression of phospho-nNOS, nNOS, RAGE, phospho-ERK, ERK, xCT, GLT-1, mGluR5, and β-tubulin blots were quantified and analyzed using ImageJ software (Version 1.53t 24). The control vehicle group was reported as 100% for determination of changes in the expression of selected target proteins in the NAc, as described previously [21,22].

### 2.6. Statistical Analyses

GraphPad Prism software (Version 10) was used to perform statistical analyses of the expression of the studied proteins. The analyses of Western blot data were conducted using one-way ANOVA followed by the Newman–Keuls post hoc multiple comparison test. The data were presented and analyzed as a percentage (relative to control values) ratio to the loading control protein, β-tubulin. The data are reported for a significance level of *p* < 0.05.

## 3. Results

### 3.1. Effects of Exposure to Hydrocodone Overdose and Ceftriaxone Treatment on Locomotion Activity

We evaluated the effects of exposure to hydrocodone overdose and ceftriaxone on locomotion activity. There was a significant difference in x activity (*n* = 7–8 mice per group, F_2,9_ = 12.97, *p* < 0.01, Figure 2A), x ambulatory (*n* = 7–8 mice per group, F_2,11_ = 8.394, *p* < 0.01, Figure 2B), and z activity (*n* = 7–8 mice per group, F_2,11_ = 37.13, *p* < 0.0001, Figure 2C) among all tested groups. The Newman–Keuls post hoc test analysis demonstrated that x activity (*p* < 0.01, Figure 2A), x ambulatory (*p* < 0.01, Figure 2B), and z activity (*p* < 0.0001, Figure 2C) significantly increased in the hydrocodone group compared to the control group. Importantly, treatment with ceftriaxone normalized x activity (*p* < 0.05, Figure 2A) and z activity (*p* < 0.001, Figure 2C) in the mice. Significant changes in locomotion activity were found between the control group and the hydrocodone–ceftriaxone group in x activity (*p* < 0.05), x ambulatory (*p* < 0.05), and z activity (*p* < 0.05) (Figure 2). However, no significant change was detected in x ambulatory between the hydrocodone and hydrocodone–ceftriaxone groups (Figure 2B).

### 3.2. Effects of Exposure to Hydrocodone Overdose and Ceftriaxone on GLT-1, xCT, and mGluR5 Protein Expressions in the NAc

The effects of exposure to hydrocodone overdose on GLT-1, xCT, and mGluR5 expression were determined in the NAc. Immunoblot analyses revealed significant differences in the expression of GLT-1 (F_2,14_ = 7.837, *p* < 0.01, Figure 3A), xCT (F_2,15_ =15.90, *p* <0.001, Figure 3B), and mGluR5 (F_2,13_ = 66.11, *p* < 0.0001, Figure 3C) in the NAc among all groups. Statistical analyses demonstrated downregulation of the expression of GLT-1 (*p* < 0.01, Figure 3A) and xCT (*p* < 0.001, Figure 3B) in the hydrocodone-treated group compared to the control group. Furthermore, exposure to hydrocodone overdose significantly increased mGluR5 expression in the NAc (*p* < 0.05, Figure 3C) in the hydrocodone group compared to the control group. Ceftriaxone treatment normalized hydrocodone-induced downregulation in GLT-1 (*p* < 0.05, Figure 3A) and xCT (*p* < 0.001, Figure 3B) expression and attenuated the effect of hydrocodone exposure on mGluR5 expression (*p* < 0.0001, Figure 3C). There were no changes in the expression of GLT-1 (Figure 3A) and xCT (Figure 3B) between the control saline and hydrocodone–ceftriaxone groups. However, there were significant changes in mGluR5 expression (*p* < 0.0001, Figure 3C) between the control and hydrocodone–ceftriaxone groups.

### 3.3. Effects of Exposure to Hydrocodone Overdose and Ceftriaxone on nNOS and ERK Protein Expression in the NAc

We next explored the effects of exposure to hydrocodone overdose and ceftriaxone on the protein expression of nNOS and ERK in the NAc. One-way ANOVA showed a significant difference in nNOS (F_2,15_ = 42.44, *p* < 0.0001, Figure 4A) and ERK (F_2,15_ = 14.10, *p* < 0.001, Figure 4B) expression among all tested groups in the NAc. Newman–Keuls post hoc analyses revealed that hydrocodone exposure upregulated nNOS expression in the NAc compared to the control group (*p* < 0.05, Figure 4A). The analysis also revealed that hydrocodone exposure downregulated ERK expression compared to the control group (*p* < 0.001, Figure 4B). Importantly, treatment with ceftriaxone significantly attenuated hydrocodone-induced upregulation of nNOS (*p* < 0.0001, Figure 4A) and hydrocodone-induced downregulation of ERK (*p* < 0.05, Figure 4B) compared to the hydrocodone group. In addition, significant differences were observed when comparing the control group with the hydrocodone–ceftriaxone group in both nNOS (*p* < 0.0001, Figure 4A) and ERK (*p* < 0.05, Figure 4B).

### 3.4. Effects of Exposure to Hydrocodone Overdose and Ceftriaxone on RAGE Protein Expression in the NAc

Lastly, we tested the effects of hydrocodone, and hydrocodone–ceftriaxone on RAGE expression. One-way ANOVA analysis demonstrated a significant difference in the expression of RAGE in the NAc among all tested groups (F_2,15_ = 4.277, *p* < 0.05, Figure 5). Newman–Keuls post hoc analysis revealed that hydrocodone exposure increased RAGE expression in the NAc compared to the control group (*p* < 0.05), and ceftriaxone treatment normalized this effect (*p* < 0.05) (Figure 5).

## 4. Discussion

In the current study, the effects of exposure to hydrocodone overdose on locomotion activity were investigated, and we determined whether ceftriaxone could modulate changes in locomotion activity in mice. Using CLAMs, we found that exposure to hydrocodone overdose increases locomotion activity in mice. In this study, we aimed to establish a hydrocodone overdose mouse model using CLAMS. We used 20 mg/kg (i.p.) of hydrocodone every other day (four i.p. injections) and then challenged the mice with a sublethal dose of hydrocodone (40 mg/kg, i.p.) since 45 mg/kg (i.p.) of hydrocodone was lethal. Thus, we investigated whether ceftriaxone attenuates the effects of sublethal hydrocodone dose in locomotor activity and the changes in the expression of target proteins. Hydrocodone-induced hyperlocomotion activity is consistent with studies demonstrating that exposure to morphine and fentanyl increases locomotor activity in rats [23,24]. Hyperlocomotion activity caused by drugs of abuse is known as locomotion sensitization. This behavioral sensitization is thought to underlie some aspects of drug dependence and is related to dopaminergic systems, which are implicated in motor function and reward [25,26]. Importantly, the present results revealed that treatment with the beta-lactam ceftriaxone, which is known to upregulate GLT-1 expression, significantly reduces hydrocodone-induced hyperlocomotion activity in mice. This is consistent with other studies showing that ceftriaxone attenuates the development of behavioral sensitization produced by chronic cocaine and amphetamine exposures [27,28]. These latter studies supported our finding (Figure 2) that ceftriaxone is associated with normalizing behavioral sensitization upon exposure to hydrocodone overdose in mice [29]. In this present study, we tested only male mice, as the aim was to establish a model of hydrocodone overdose using CLAMS and to determine whether ceftriaxone attenuates the effect of hydrocodone exposure, particularly with a higher dose. Further studies are warranted to investigate the effects of extended duration of exposure to opioids in the brains of male as well as female mice for determination of sex difference. In addition, further studies are warranted to determine the effects of different doses of hydrocodone and ceftriaxone, as well as to test a novel beta-lactam non-antibiotic such as MC-100093, which has been shown to be protective in the brain of rats exposed to ethanol [22].

Increased extracellular glutamate concentrations at the synaptic cleft can lead to glutamate neuroexcitotoxicity, which might be associated with certain neuroinflammatory and neurodegenerative diseases [30,31,32]. Therefore, maintaining glutamate homeostasis is very important. GLT-1 and xCT are highly expressed in astrocytes and help remove excess extracellular glutamate concentrations from the synaptic cleft. Hydrocodone exposure has been associated with reduced GLT-1 and xCT, resulting in an elevation of extracellular glutamate concentrations in the NAc [28]. Previous studies from our laboratory and others have demonstrated that a reduction in GLT-1 expression in the brain is associated with chronic exposure to substances of abuse [33,34,35]. Furthermore, we recently reported that chronic hydrocodone exposure induces downregulation of GLT-1 and xCT expression in the mesocorticolimbic brain region, and that ceftriaxone treatment attenuates hydrocodone-induced downregulation of GLT-1 and xCT expression [16]. These findings are consistent with our current results showing downregulation of GLT-1 and xCT in the NAc following exposure to hydrocodone overdose. Importantly, ceftriaxone treatment restored GLT-1 and xCT expression in the NAc. Therefore, the current and previous findings suggest that chronic exposure to hydrocodone overdose may lead to dysregulation of glutamate homeostasis in the brain and that this effect can be attenuated by ceftriaxone treatment. Alternatively, we investigated the effects of hydrocodone and ceftriaxone on the expression of mGluR5 in the NAc. mGluR5 is highly expressed in the brain and is involved in mediating the potentiating effects of opioids [36,37,38]. Previous studies from our laboratory and others have revealed an increase in mGluR5 expression in the mesocorticolimbic brain regions during chronic exposure to hydrocodone and in morphine place preference paradigms [16,39]. Our results showed that mGluR5 increased in the NAc following exposure to hydrocodone overdose. Hydrocodone-induced upregulation of mGluR5 expression was attenuated with ceftriaxone treatment. Glutamate acts on mGluR, which is coupled to intracellular second messengers via G proteins, guanine nucleotide regulatory, or phosphorylation of MAP kinase [40,41]. Activation of mGluR5 may result in cellular depolarization and increased neuronal excitability. mGluR5 is positively coupled to phosphatidylinositol (PI) hydrolysis, leading to the activation of protein kinase C and increasing intracellular calcium ions Ca^2+^ [42,43,44]. The increase in intracellular calcium ions may induce the production of nitric oxide (NO) through Ca^2+^/calmodulin activation of nNOS, and high NO concentration can trigger numerous downstream neurotoxic cascades. It has been shown in studies by others and us that nNOS activity (phosphorylated nNOS resulting in a higher p-nNOS/nNOS ratio) increases following cue-induced reinstatement of amphetamine, cocaine seeking, and chronic ethanol exposure in the NAc of mice and rats [45,46,47]. These studies support our finding that the *p*-nNOS/nNOS ratio increased in the NAc after exposure to hydrocodone overdose. Importantly, ceftriaxone treatment reversed the effects of hydrocodone-induced upregulation of the *p*-nNOS/nNOS ratio.

This study demonstrated that exposure to hydrocodone overdose is accompanied by a decreased phosphorylation of ERK in the NAc and that this effect was attenuated with ceftriaxone treatment. This is consistent with previous findings from our laboratory showing that ceftriaxone attenuates hydrocodone-induced downregulation of *p*-ERK expression in the mesocorticolimbic brain regions [16]. ERK is involved in the regulation of GLT-1 transcription through the initiation of nuclear transcription factor-κβ (NF-κβ) and cAMP response element-binding protein (CREB). Notably, nNOS-derived NO can also regulate synaptic plasticity by inducing the ERK signaling pathway [48]. A prior study indicated that the inhibition of the ERK signaling pathway is due to the generation of free radicals upon the activation of nNOS in vitro [49]. These studies supported our findings that downregulation of GLT-1 expression is associated with increased nNOS activity and decreased ERK expression in the NAc of mice exposed to hydrocodone overdose.

Furthermore, this study investigated the RAGE signaling pathways with exposure to hydrocodone overdose and ceftriaxone treatment. The RAGE is known to induce neuroinflammation through activation of the NF-κB signaling pathway [50,51,52]. Our current analysis showed that exposure to hydrocodone overdose increased RAGE expression in the NAc, indicating the role of inflammatory factors in opioid overdose events and further validating the induction of brain inflammation. Interestingly, ceftriaxone attenuated hydrocodone-induced increases in RAGE expression. Studies from other laboratories confirmed our findings, demonstrating that pharmacological inhibition of RAGE attenuated neuroinflammation in the brain [53,54,55].

We propose here that activation of the mGluR5-nNOS-ERK pathway reduces GLT-1 expression, leading to excessive extracellular glutamate concentrations in the brain, thereby increasing neuroexcitotoxicity (Figure 6). Glutamate also binds to the N-methyl-d-aspartate receptor (NMDAR) and activates the enzyme nNOS to produce NO. Increased NO expression inhibits ERK production and inactivates the downstream signaling pathway of ERK (Figure 6). Furthermore, hydrocodone upregulates RAGE expression, leading to neuroinflammation.

## 5. Conclusions

Exposure to hydrocodone overdose induces hyperlocomotion activity in mice. Ceftriaxone treatment successfully attenuates hydrocodone-induced hyperlocomotion activity. In addition, exposure to hydrocodone overdose decreases GLT-1 and xCT expression in the NAc, thereby disrupting glutamate homeostasis. Increased extracellular glutamate concentrations at the synaptic cleft may overstimulate mGluR5 and increase nNOS activity. As observed in this study, activation of nNOS activity can lead to inhibition of the ERK signaling pathway. Furthermore, exposure to hydrocodone overdose increases RAGE expression, thereby inducing neuroinflammation in the brain. However, ceftriaxone treatment attenuates hydrocodone-induced upregulation of mGluR5, NOS activity, and RAGE, as well as hydrocodone-induced downregulation of GLT-1, xCT, and ERK expression. Future studies are warranted to investigate the beneficial effects of other novel synthetic beta-lactams (non-antibiotics), and longer exposure of doses of hydrocodone on opioid-induced hyperlocomotion activity, dysregulation of glutamatergic systems, and neuroinflammation.

## Figures and Tables

**Figure 1 brainsci-14-00361-f001:**
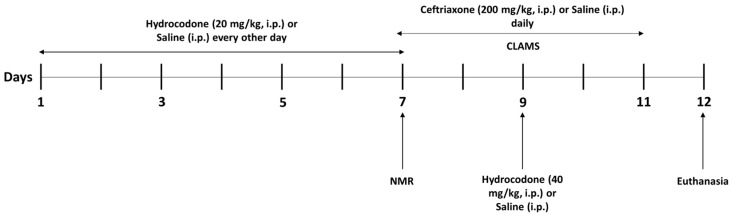
Timeline of the experimental procedure. CLAMS, comprehensive laboratory animal monitoring system.

**Figure 2 brainsci-14-00361-f002:**
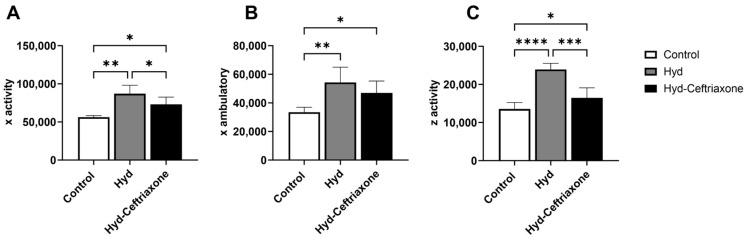
Effects of exposure to hydrocodone overdose on locomotion activity in mice. (**A**) Statistical analysis revealed that x activity increased in the hydrocodone-treated group compared to the control group, while treatment with ceftriaxone (200 mg/kg, i.p.) reduced x activity compared to the hydrocodone-treated group. (**B**) Statistical analysis demonstrated that x ambulatory increased in the hydrocodone-treated group compared to the control group, and there was no significant difference in x ambulatory in the hydrocodone–ceftriaxone group compared to the hydrocodone-treated group. (**C**) Statistical analysis revealed that z activity increased in the hydrocodone-treated group compared to the control group, and ceftriaxone treatment (200 mg/kg, i.p.) reduced z activity compared to the hydrocodone-treated group. Data from the control group are represented as 100%. Each column is expressed as mean ± S.E.M (*n* = 7–8/group), (* *p* < 0.05, ** *p* < 0.01, *** *p* < 0.001 and **** *p* < 0.0001). Hyd, hydrocodone.

**Figure 3 brainsci-14-00361-f003:**
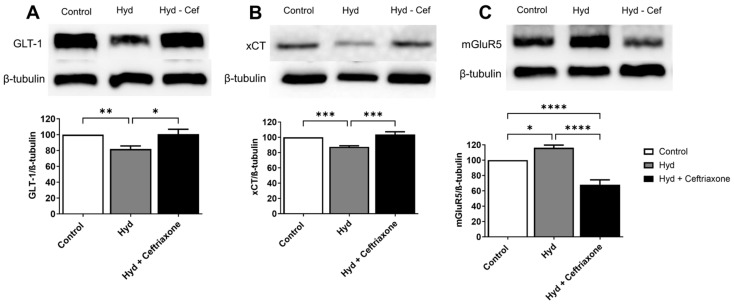
Effects of exposure to hydrocodone overdose on GLT-1, xCT, and mGluR5 expression in the NAc. (**A**) Western blots for GLT-1 and β-tubulin in the NAc. One-way ANOVA followed by the Newman–Keuls multiple comparisons test showed downregulation of GLT-1 expression in the hydrocodone-treated group compared to the control group, and ceftriaxone treatment (200 mg/kg, i.p.) normalized GLT-1 expression in the NAc compared to the hydrocodone-treated group. (**B**) One-way ANOVA followed by the Newman–Keuls multiple comparisons test revealed downregulation of xCT expression in the hydrocodone-treated group compared to the control group, and ceftriaxone treatment (200 mg/kg, i.p.) normalized xCT expression in the NAc compared to the hydrocodone-treated group. (**C**) One-way ANOVA followed by the Newman–Keuls multiple comparisons test revealed upregulation of mGluR5 expression in the hydrocodone-treated group compared to the control group, and ceftriaxone treatment (200 mg/kg, i.p.) attenuated this effect. There was also a significant difference between the control and hydrocodone–ceftriaxone-treated groups in the expression of mGluR5 in the NAc. Data from the control group are represented as 100%. Each column is expressed as mean ± S.E.M (*n* = 7–8/group), (* *p* < 0.05, ** *p* < 0.01, *** *p* < 0.001 and **** *p* < 0.0001). GLT-1, glutamate transporter 1; xCT, cystine/glutamate antiporter; mGluR5, metabotropic glutamate receptor subtype 5; Hyd, hydrocodone.

**Figure 4 brainsci-14-00361-f004:**
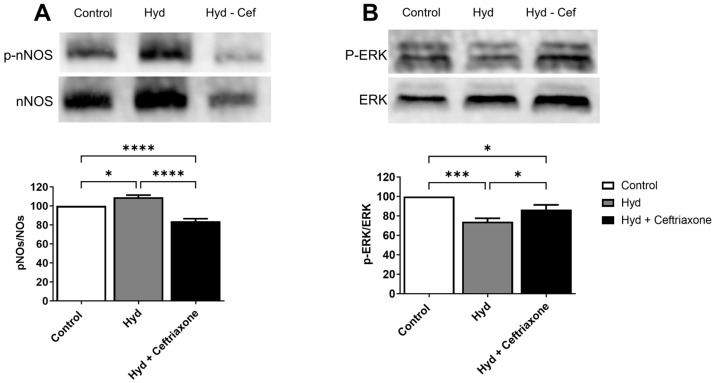
Effects of exposure to hydrocodone overdose on the expression of nNOS and ERK in the NAc. (**A**) Western blots for p-nNOS and NOS in the NAc. One-way ANOVA followed by the Newman–Keuls multiple comparisons test revealed that hydrocodone exposure increased nNOS expression compared to the control group, and ceftriaxone treatment (200 mg/kg, i.p.) decreased nNOS expression in the NAc compared to the hydrocodone group. (**B**) Western blots for p-ERK and ERK in the NAc. One-way ANOVA followed by the Newman–Keuls multiple comparisons test revealed that hydrocodone exposure downregulated ERK expression compared to the control group, and ceftriaxone treatment (200 mg/kg, i.p.) upregulated ERK expression in the NAc compared to the hydrocodone group. Data from the control group are represented as 100%. Each column is expressed as mean ± S.E.M (*n* = 7–8/group), (* *p* < 0.05, *** *p* < 0.001 and **** *p* < 0.0001). nNOS, neuronal nitric oxide synthase; ERK, extracellular signal-regulated kinases; Hyd, hydrocodone.

**Figure 5 brainsci-14-00361-f005:**
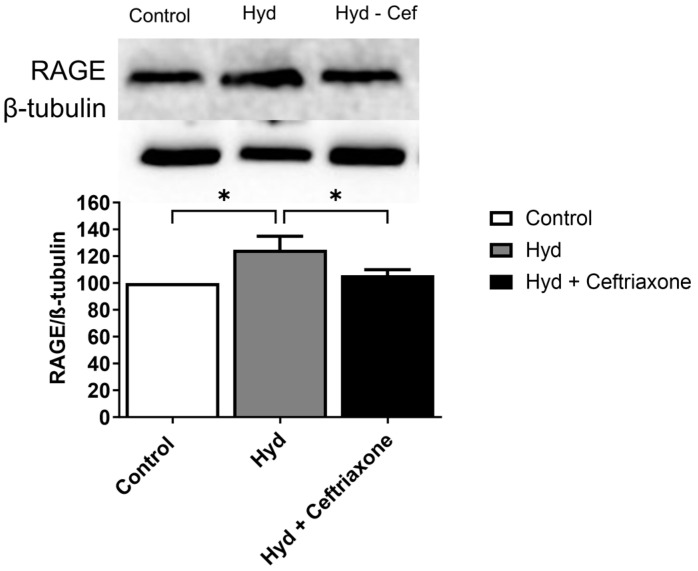
Effects of exposure to hydrocodone overdose on RAGE expression in the NAc. One-way ANOVA followed by the Newman–Keuls multiple comparisons test showed that RAGE expression was upregulated in the hydrocodone group compared to the control group, while ceftriaxone (200 mg/kg) downregulated RAGE expression in the NAc compared to the hydrocodone group. Data from the control group are represented as 100%. Each column is expressed as mean ± S.E.M (*n* = 7–8/group), (* *p* < 0.05). RAGE, receptor for advanced glycation end products; Hyd, hydrocodone.

**Figure 6 brainsci-14-00361-f006:**
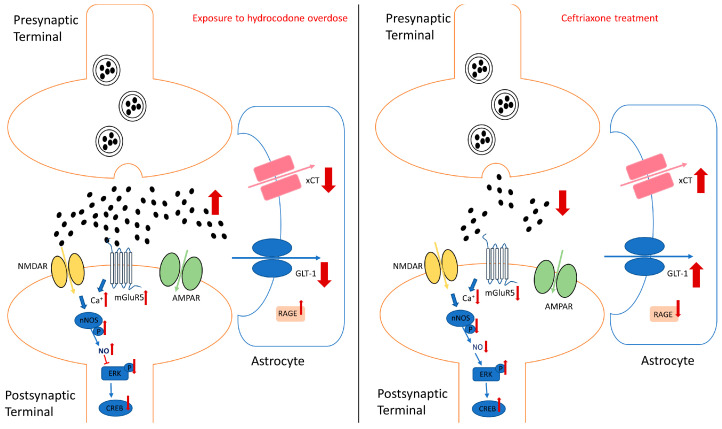
Schematic representation summarizing the effects of exposure to hydrocodone overdose on the mGluR5-nNOS-ERK pathway and GLT-1, xCT, and RAGE expression in the NAc. Exposure to hydrocodone overdose may increase synaptic glutamate release, resulting in an increase in extracellular glutamate concentrations. Under a hyper-glutamatergic state, mGluR5 and NMDAR are overstimulated, thereby increasing intracellular calcium and subsequently upregulating nNOS activity. Activation of nNOS activity can lead to inhibition of the downstream ERK signaling pathway. Additionally, exposure to hydrocodone overdose is associated with an increase in the inflammatory response, such as upregulation of RAGE. Ceftriaxone treatment attenuates hydrocodone-induced mGluR5-nNOS-ERK pathway activation, glutamatergic system dysregulation, and RAGE upregulation. (Blue arrows indicate the downstream pathways; Red arrows indicate upregulation or downregulation of all target proteins or markers).

## Data Availability

The data are contained within this article.

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
