# Peer review of "Effects of Hydrocodone Overdose and Ceftriaxone on Astrocytic Glutamate Transporters and Glutamate Receptors, and Associated Signaling in Nucleus Accumbens as well as Locomotor Activity in C57/BL Mice"

_brainsci, 2024, doi:10.3390/brainsci14040361_

Round 1

Reviewer 1 Report

Comments and Suggestions for Authors

The primary research question investigates the effects of escalated doses of hydrocodone on locomotor activity, expression of target proteins related to the glutamatergic system, signaling kinases, and neuroinflammatory factors in the nucleus accumbens of mice. Additionally, it examines the therapeutic potential of ceftriaxone in modulating these effects.

As a reviewer, here are my suggestions for the Authors regarding the methodology and related discussion in the manuscript:

1.       Please provide a detailed acknowledgment of the methodological choices made in the study. This should include a justification for the selection of C57BL/6 mice, the specific doses of hydrocodone and ceftriaxone, and the rationale behind the experimental conditions and durations. Clarifying these choices will help readers understand the context and limitations of the study.

2.       It would be beneficial to include a section discussing the limitations of the current methodology. Specifically, address the use of only male mice and the implications of this for the generalizability of the results. Additionally, suggest directions for future research, such as varying the drug dosages, extending the duration of treatment, or including female mice to explore potential sex differences in response.

3.       While it's noted that the study was approved by the Institutional Animal Care and Use Committee (IACUC) and follows NIH guidelines, expanding on the ethical considerations and animal welfare measures taken during the study would enhance the transparency and ethical rigor of the manuscript.

Author Response

We would like to thank the reviewers for their valuable comments and insights to improve the quality of the manuscript. We have addressed point-by-point the reviews’ comments. Changes and addressed comments can found in bold below in the responses to the reviewers in this document and text of the manuscript.

Reviewer #1:

As a reviewer, here are my suggestions for the Authors regarding the methodology and related discussion in the manuscript:

  1. Please provide a detailed acknowledgment of the methodological choices made in the study. This should include a justification for the selection of C57BL/6 mice, the specific doses of hydrocodone and ceftriaxone, and the rationale behind the experimental conditions and durations. Clarifying these choices will help readers understand the context and limitations of the study.

- We have included the following statement in the method section under Animal and Study Design section to address the reviewer’s comment: “C57BL/6 mice were used in this study. These mice were reported to show robust differences in drug dependence-relevant behaviors, including locomotor sensitization to substance of abuse (Kirkpatrick and Bryant, 2015; Orsini et al., 2005). ”

Furthermore, we have included the following in the discussion section: “Previous study from our laboratory used 10 mg/kg hydrocodone i.p. injection daily and 200 mg/kg of ceftriaxone treatment to determine the chronic effect of hydrocodone in mice (Wong and Sari, 2023).  In this study, we increased the dosage to 20 mg/kg hydrocodone i.p. injection every other day.  We also challenged the mice with the highest dose of 40 mg/kg to determine the effects of a higher dose of hydrocodone in mouse brains. We have chosen these dosages to mimic the escalated regimen occurring in human with opioids misuse.

  1. It would be beneficial to include a section discussing the limitations of the current methodology. Specifically, address the use of only male mice and the implications of this for the generalizability of the results. Additionally, suggest directions for future research, such as varying the drug dosages, extending the duration of treatment, or including female mice to explore potential sex differences in response.

- We would like to thank the reviewer for these suggestions. We have added the following in the discussion section: “In this study, we tested only male mice as the aim was to establish a model of escalated doses of hydrocodone using CLAMS, and to determine whether ceftriaxone would attenuate the effect of hydrocodone exposure, particularly with higher dose.  Further studies are warranted to investigate the effects of extended duration of exposure to opioids in brains of male as well as female mice for determination of sex difference.  In addition, further studies are warranted to determine the effects of different doses of hydrocodone and ceftriaxone as well as testing a novel beta-lactam non-antibiotic such as MC-100093, which has been shown to be protective in the brain of rats exposed to ethanol (Alhaddad et al., 2022).”

  1. While it's noted that the study was approved by the Institutional Animal Care and Use Committee (IACUC) and follows NIH guidelines, expanding on the ethical considerations and animal welfare measures taken during the study would enhance the transparency and ethical rigor of the manuscript.

We have added the following as per the reviewer’s comment: “Mice were i.p. injected and handled with care to prevent any distress and minimize any pain.  Mice were monitored everyday throughout the study, particularly when hydrocodone treatment was performed, and this was to determine any potential health issues.  Mice were euthanized using C02 inhalation at the end of the experiments.” 

References:

Alhaddad, H., et al., 2022. Effects of a Novel Beta Lactam Compound, MC-100093, on the Expression of Glutamate Transporters/Receptors and Ethanol Drinking Behavior of Alcohol-Preferring Rats. J Pharmacol Exp Ther. 383, 208-216.

Kirkpatrick, S.L., Bryant, C.D., 2015. Behavioral architecture of opioid reward and aversion in C57BL/6 substrains. Frontiers in behavioral neuroscience. 8, 450.

Orsini, C., et al., 2005. Susceptibility to conditioned place preference induced by addictive drugs in mice of the C57BL/6 and DBA/2 inbred strains. Psychopharmacology. 181, 327-336.

Wong, W., Sari, Y., 2023. Effects of Chronic Hydrocodone Exposure and Ceftriaxone on the Expression of Astrocytic Glutamate Transporters in Mesocorticolimbic Brain Regions of C57/BL Mice. Toxics. 11, 870.

Reviewer 2 Report

Comments and Suggestions for Authors

The reviewer would like to declare no conflict of interest with the authors and their affiliated institutions.

The authors presented a rather interesting study, but some fundamental ideas will need to be clarified before the manuscript can be accepted for publication.

1. Line 49-50, the authors stated that the effect of hydrocodone on locomotor sensitization was not studied previously. However, this statement is not correct. Kindly refer to PMID 21600913 and 25617530.

2. The authors repeatedly emphasised that they the study is focusing on mice treated with escalated dose of hydrocodone. What is the justification of choosing this dosing regime? what is the "baseline" effects of non-escalated dose treatment of hydrocodone? The authors did not show the effect of escalated vs non-escalated doses on NAcc glutamatergic signalings.

3. In clinical setting, patients who receive chronic opioid treatment require gradual increase in opioid dose, due to the development of tolerance to opioid treatment. In this study, 4 injections of hydrocodone in a span of 7 days, would it cause tolerance? if no, then what is the justification of inclding the dose to double in the 5th injection?

4. Newman Keuls test was chosen as the post hoc test for One Way ANOVA in this study. Kindly justify why this test was chosen.

Author Response

We would like to thank the reviewers for their valuable comments and insights to improve the quality of the manuscript. We have addressed point-by-point the reviews’ comments. Changes and addressed comments can found in bold below in the responses to the reviewers in this document and text of the manuscript.

Reviewer#2:

The reviewer would like to declare no conflict of interest with the authors and their affiliated institutions.

The authors presented a rather interesting study, but some fundamental ideas will need to be clarified before the manuscript can be accepted for publication.

  1. Line 49-50, the authors stated that the effect of hydrocodone on locomotor sensitization was not studied previously. However, this statement is not correct. Kindly refer to PMID 21600913 and 25617530.

- We thank the reviewer for this informative suggestion. We have modified this statement accordingly and added the suggested references: “Other studies showed that exposure to opioids, including hydrocodone, induced locomotor sensitization (Emery et al., 2015; Nazarian et al., 2011).  Although, previous study from our laboratory showed that ceftriaxone, beta-lactam antibiotic known to upregulate major glutamate transporter, GLT-1, attenuated reinstatement to hydrocodone using conditioned place preference (Alshehri et al., 2018), the effect of ceftriaxone in attenuating hydrocodone-induced locomotor sensitization is less studied.”

  1. The authors repeatedly emphasised that they the study is focusing on mice treated with escalated dose of hydrocodone. What is the justification of choosing this dosing regime? what is the "baseline" effects of non-escalated dose treatment of hydrocodone? The authors did not show the effect of escalated vs non-escalated doses on NAcc glutamatergic signalings.

We thank the reviewer for this comment: We have added the following to address the reviewer’s comment: “Previous study from our laboratory used 10 mg/kg hydrocodone i.p. injection daily and 200 mg/kg of ceftriaxone treatment to determine the chronic effect of hydrocodone in mice (Wong and Sari, 2023).  In this study, we increased the dosage to 20 mg/kg hydrocodone i.p. injection every other day.  We also challenged the mice with the highest dose of 40 mg/kg to determine the effects of a higher dose of hydrocodone in mouse brains. We have chosen these dosages to mimic the escalated regimen occurring in human with opioids misuse.”

  1. In clinical setting, patients who receive chronic opioid treatment require gradual increase in opioid dose, due to the development of tolerance to opioid treatment. In this study, 4 injections of hydrocodone in a span of 7 days, would it cause tolerance? if no, then what is the justification of inclding the dose to double in the 5th injection?

- Previous study from our lab used four injection showed development of tolerance using conditioned place preference (Alshehri et al., 2018).  This study used higher lower dose repeated exposure and higher dose to mimic the overdose in clinical setting and determine whether ceftriaxone would attenuate the effect of this regimen with focus on target proteins and locomotor activity. We have discussed the rationale for testing the dosing of hydrocodone as well as the use of ceftriaxone.

  1. Newman Keuls test was chosen as the post hoc test for One Way ANOVA in this study. Kindly justify why this test was chosen.

- This is an important point raised by the reviewer.  The Newman-Keuls post hoc test uses different critical values to compare pairs of means. Additionally, Newman-Keuls can be safely used when comparing up to three sets of means.  Since we are comparing three groups in this study, it is accurate to use Newman-Keuls as a post hoc test.  Furthermore, our laboratory consistently used the Newman-Keuls post hoc test for all the past studies published in several peer reviewed high impact journals.

References:

Alshehri, F.S., et al., 2018. Effects of ceftriaxone on hydrocodone seeking behavior and glial glutamate transporters in P rats. Behavioural brain research. 347, 368-376.

Emery, M.A., et al., 2015. Differential effects of oxycodone, hydrocodone, and morphine on the responses of D2/D3 dopamine receptors. Behav Brain Res. 284, 37-41.

Nazarian, A., Are, D., Tenayuca, J.M., 2011. Acetaminophen modulation of hydrocodone reward in rats. Pharmacol Biochem Behav. 99, 307-10.

Wong, W., Sari, Y., 2023. Effects of Chronic Hydrocodone Exposure and Ceftriaxone on the Expression of Astrocytic Glutamate Transporters in Mesocorticolimbic Brain Regions of C57/BL Mice. Toxics. 11, 870.

Round 2

Reviewer 1 Report

Comments and Suggestions for Authors

I would like to thank the Authors for the improvements they have done. I have no further comments.

Author Response

Thank you to the reviewer for the time and effort.

Reviewer 2 Report

Comments and Suggestions for Authors

The rebuttal points by the authors do not improve the manuscript.

Rebuttal to point #3

"- Previous study from our lab used four injection showed development of tolerance using conditioned place preference (Alshehri et al., 2018). This study used higher lower dose repeated exposure and higher dose to mimic the overdose in clinical setting and determine whether ceftriaxone would attenuate the effect of this regimen with focus on target proteins and locomotor activity. We have discussed the rationale for testing the dosing of hydrocodone as well as the use of ceftriaxone."

Kindly take note of the difference between "tolerance" and "dependance". From my humble experience, CPP does not measure tolerance, it is a behavioral test indicative of possible drug dependance behaviors. Thus the author failed to address this point. 

Rebuttal point #2

"We thank the reviewer for this comment: We have added the following to address the reviewer’s comment: “Previous study from our laboratory used 10 mg/kg hydrocodone i.p. injection daily and 200 mg/kg of ceftriaxone treatment to determine the chronic effect of hydrocodone in mice (Wong and Sari, 2023). In this study, we increased the dosage to 20 mg/kg hydrocodone i.p. injection every other day. We also challenged the mice with the highest dose of 40 mg/kg to determine the effects of a higher dose of hydrocodone in mouse brains. We have chosen these dosages to mimic the escalated regimen occurring in human with opioids misuse.”"

Again, the author did not directly address the question. 1. the current treatment protocol does not mimic escalated doses treatment in clinical setting, 2. there's no valid justification why such design is developed. The repeated 10mg/kg dose experiment was not directly compared with the current study.

Another concern is the high similarity index detected by iThenticate. The authors need to address this or provide an explanation. 

Author Response

We would like to thank the reviewers for their valuable comments and insights to improve the quality of the manuscript. We have addressed point-by-point the reviews’ comments. Changes and addressed comments can found in bold below in the responses to the reviewers in this document and text of the manuscript.

Reviewer #2 comments:

The rebuttal points by the authors do not improve the manuscript.

Rebuttal to point #3

"- Previous study from our lab used four injection showed development of tolerance using conditioned place preference (Alshehri et al., 2018). This study used higher lower dose repeated exposure and higher dose to mimic the overdose in clinical setting and determine whether ceftriaxone would attenuate the effect of this regimen with focus on target proteins and locomotor activity. We have discussed the rationale for testing the dosing of hydrocodone as well as the use of ceftriaxone."

Kindly take note of the difference between "tolerance" and "dependance". From my humble experience, CPP does not measure tolerance, it is a behavioral test indicative of possible drug dependance behaviors. Thus the author failed to address this point. 

-           We apologize for the confusion. We agree with the reviewer that CPP doesn’t measure tolerance. In our previous response, we meant the effect of hydrocodone exposure in conditioned place preference and the effect of ceftriaxone on reinstatement to hydrocodone. We have checked the manuscript again and the term tolerance and dependence are omitted. The purpose of our present study was to establish a hydrocodone overdose model using the Comprehensive Laboratory Animal Monitoring System (CLAMS). In the current study, we used ceftriaxone (a beta-lactam antibiotic) as a treatment to see if ceftriaxone could reverse the adverse effects of hydrocodone overdose. Future studies warrant determining the effects of other nonantibiotic β-lactams (MC-100093) in mouse models of overdose using CLAMS.

The following statement has been revised in the introduction as per the reviewer’s comment:

Other studies showed that exposure to opioids, including hydrocodone, induced locomotor sensitization [1, 2].  Although, previous study from our laboratory showed that ceftriaxone, beta-lactam antibiotic known to upregulate major glutamate transporter, GLT-1, attenuated reinstatement to hydrocodone using conditioned place preference [3], the effect of ceftriaxone in attenuating hydrocodone-induced locomotor sensitization is less studied.  Thus, in this study, we used the Comprehensive Laboratory Animal Monitoring System (CLAMS) to determine the locomotion activity in mice exposed to repeated doses of hydrocodone with challenge with higher dose, and ceftriaxone treatment by breaking the infrared beam using infrared photocell technology.”

Rebuttal point #2

"We thank the reviewer for this comment: We have added the following to address the reviewer’s comment: “Previous study from our laboratory used 10 mg/kg hydrocodone i.p. injection daily and 200 mg/kg of ceftriaxone treatment to determine the chronic effect of hydrocodone in mice (Wong and Sari, 2023). In this study, we increased the dosage to 20 mg/kg hydrocodone i.p. injection every other day. We also challenged the mice with the highest dose of 40 mg/kg to determine the effects of a higher dose of hydrocodone in mouse brains. We have chosen these dosages to mimic the escalated regimen occurring in human with opioids misuse.”"

Again, the author did not directly address the question. 1. the current treatment protocol does not mimic escalated doses treatment in clinical setting, 2. there's no valid justification why such design is developed. The repeated 10mg/kg dose experiment was not directly compared with the current study.

We apologize the reviewer for the lack of clarity and missing information about the use of higher dose. We have changed the statement in the discussion to address the reviewer’s comments. The following statement is in the discussion section: “In this study, we aimed to establish a hydrocodone overdose mouse model using CLAMS. We used 20 mg/kg (i.p.) of hydrocodone every other day (four i.p. injections) and then challenged the mice with a sublethal dose of hydrocodone (40 mg/kg, i.p.) since 45 mg/kg (i.p.) of hydrocodone was lethal. Thus, we investigated on whether ceftriaxone would attenuate the effects of sublethal hydrocodone dose in locomotor activity and the changes in the expression of target proteins.”

We have also added the following statement in the method section: “It is important to note that 40 mg/kg (i.p.) of hydrocodone was considered as sublethal dose since we found that 45 mg/kg (i.p.) of hydrocodone was lethal in mice (n=3). Thus, we have chosen to test 40 mg/kg (i.p.) of hydrocodone as higher and sublethal dose.”

We are aiming in the future studies to test other beta-lactams such as MC-100093 [4] to study other adverse effects, such as respiratory depression, using extended hydrocodone exposure regimens as well as CLAMS.

In addition, we have tune down the statements or wording related to clinic and human use of opioids.

Another concern is the high similarity index detected by iThenticate. The authors need to address this or provide an explanation. 

We understand the concern of the reviewer regarding the similarity index, and we have tried to our best to reduce the similarity.  However, after reviewing the manuscript, we found that most of word similarities are just in the methods section (as the protocol for samples dissection and western blotting is standard in our laboratory), author contribution, funding, Institutional review board statement, acknowledgment, with some standard terms such as chronic exposure, mesocorticolimbic brain region, and others, and text cited by citations. There are many keywords for protein, groups, and others that we couldn’t change. We would be willing to change anything that the reviewer and editor may recommend.

References

  1. Emery, M.A., et al., Differential effects of oxycodone, hydrocodone, and morphine on the responses of D2/D3 dopamine receptors. Behav Brain Res, 2015. 284: p. 37-41.
  2. Nazarian, A., D. Are, and J.M. Tenayuca, Acetaminophen modulation of hydrocodone reward in rats. Pharmacol Biochem Behav, 2011. 99(3): p. 307-10.
  3. Alshehri, F.S., et al., Effects of ceftriaxone on hydrocodone seeking behavior and glial glutamate transporters in P rats. Behavioural brain research, 2018. 347: p. 368-376.
  4. Alhaddad, H., et al., Effects of a Novel Beta Lactam Compound, MC-100093, on the Expression of Glutamate Transporters/Receptors and Ethanol Drinking Behavior of Alcohol-Preferring Rats. Journal of Pharmacology and Experimental Therapeutics, 2022. 383(3): p. 208-216.

Round 3

Reviewer 2 Report

Comments and Suggestions for Authors

Noted that the authors has made amendments to the flow of the study.  Since the author stated in the discussion that "In this study, we aimed to establish a hydrocodone overdose mouse model using CLAMS....", the title, abstract, introduction MUST be adjusted along with this change. The title and abstract still stated "escalated dose of hydrocodone"...

Author Response

We thank the reviewer for this important comment. We have  adjusted the working to "hydrocodone overdose" in the title, abstract, introduction and other sections in the manuscript. Changes are highlighted in bold in the manuscript.